# Breaking Bad Proteins—Discovery Approaches and the Road to Clinic for Degraders

**DOI:** 10.3390/cells13070578

**Published:** 2024-03-26

**Authors:** Corentin Bouvier, Rachel Lawrence, Francesca Cavallo, Wendy Xolalpa, Allan Jordan, Roland Hjerpe, Manuel S. Rodriguez

**Affiliations:** 1Laboratoire de Chimie de Coordination LCC-UPR 8241-CNRS, 31077 Toulouse, France; corentin.bouvier@lcc-toulouse.fr (C.B.); manuel.rodriguez@lcc-toulouse.fr (M.S.R.); 2Sygnature Discovery, Bio City, Pennyfoot St., Nottingham NG1 1GR, UKf.cavallo@sygnaturediscovery.com (F.C.); a.jordan@sygnaturediscovery.com (A.J.); 3Departamento de Ingeniería Celular y Biocatálisis, Instituto de Biotecnología, Universidad Nacional Autónoma de México, Cuernavaca 62209, Morelos, Mexico; wendy.xolalpa@ibt.unam.mx; 4Pharmadev, UMR 152, Université de Toulouse, IRD, UT3, 31400 Toulouse, France; 5B Molecular, Centre Pierre Potier, Canceropôle, 31106 Toulouse, France

**Keywords:** TPD, PROTAC, molecular glue degrader, heterobifunctional degrader, ubiquitin, proteasome, targeted protein degradation

## Abstract

Proteolysis-targeting chimeras (PROTACs) describe compounds that bind to and induce degradation of a target by simultaneously binding to a ubiquitin ligase. More generally referred to as bifunctional degraders, PROTACs have led the way in the field of targeted protein degradation (TPD), with several compounds currently undergoing clinical testing. Alongside bifunctional degraders, single-moiety compounds, or molecular glue degraders (MGDs), are increasingly being considered as a viable approach for development of therapeutics, driven by advances in rational discovery approaches. This review focuses on drug discovery with respect to bifunctional and molecular glue degraders within the ubiquitin proteasome system, including analysis of mechanistic concepts and discovery approaches, with an overview of current clinical and pre-clinical degrader status in oncology, neurodegenerative and inflammatory disease.

## 1. General Concepts in TPD Drug Discovery

The concept of pharmacologically modulating turnover of a specific target protein by leveraging the ubiquitin–proteasome system (UPS) was first realized by Ray Deshaies, Craig Crews and others in a landmark study from 2001 [1], where the authors coined the term proteolysis-targeting chimeras (PROTACs). PROTACs, or more formally, bifunctional degraders, are composed of two separate moieties, a ubiquitin ligase (also called E3) recruiter and a target recruiter, with a linker connecting the two. In this way, bifunctional degraders can bind to the ligase and target at the same time, effectively putting the ligase into close proximity with the protein of interest (POI), leading to its ubiquitination and proteasomal proteolysis (Figure 1A).

Molecular glue degraders (MGDs), in contrast, consist of a single moiety and have a conceptually distinct mode of generating the ternary complex. Whereas bifunctional degraders rely on two separate ligands for ligase and POI recruitment, and can independently and separately bind both, glue degraders must first bind to one of the proteins, which modulates the surface properties of the bound protein to induce binding to the second. Most reported MGDs are ligase ligands, where compound binding induces interaction with a neo-substrate, a protein that otherwise would not interact with or be degraded by the ligase. An MGD can also bind the target protein, with resulting induced binding to a ligase, as exemplified by degraders for Cyclin K [2,3,4,5] and BRD4 [6,7,8]. In addition to their therapeutic potential, degraders represent valuable tools for exploration of fundamental cell biology, where the functional consequences of acute protein ablation can provide novel insights into cell physiology and the intricate regulatory networks that govern cellular functions. 

Due to being composed of two ligands and a linker, the molecular weight and physicochemical properties of bifunctional degraders are outside of the rule of five [9], which poses significant challenges in developing these compounds for an oral route of administration. These challenges include optimization of compounds to reduce, e.g., molecular weight (MW), number of hydrogen bond donors (HBD), rotatable bonds (RB), and topological polar surface area (TPSA), whilst maintaining potency. In addition, heterobifunctional degraders may behave unpredictably, where changes in compound structure impact on the conformations that the compound adopts in different environments, a property described as molecular chameleonicity [10,11]. Guidelines for optimization of PROTAC physicochemical properties with respect to oral bioavailability have recently been published by Arvinas, recommending, among other properties, MW < 950, HBD < 2, RB < 14 and TPSA < 200 Å^2^ [12]. 

In contrast to PROTACs, molecular glue degraders are generally compact, with a low molecular weight and the potential for favorable drug-like physiochemical properties, simplifying the late-stage optimization process. The challenge with MGDs lies largely in the identification of compounds, with the rational identification of glues between a selected ligase-target pair remaining a significant challenge. 

Both MGDs and bifunctional degraders have event-driven pharmacology. In contrast, conventional small-molecule inhibitors function by occupying a binding pocket on the target protein, for example an enzyme active site. Degraders circumvent this requirement by acting catalytically, where a single degrader molecule turns over multiple target protein molecules in a substoichiometric mode [13]. This event-driven mechanism of action is different from occupancy-based inhibitors, where the catalytic property may allow lower or less frequent dosing.

In this context, it is worth noting that degraders may be developed as covalent compounds, and interestingly the first PROTAC was based on the covalent target ligand ovalicin [1]. Since the catalytic advantage is lost for degraders that covalently bind the target, covalent bifunctional degraders have largely been explored for ligands reacting with the ligase [14,15,16,17]. Importantly, covalent E3 recruiters cannot be inhibitory of ligase function (see, e.g., covalent ligands for the E3 HOIP described by Johanssen et al. [18]) since this would invalidate their utility in a TPD setting, and thus avoiding reaction with active site residues in ligases that have enzymatic activity is essential. The concept of target-covalent degraders has been explored for BTK, utilizing both reversible and irreversible covalent ibrutinib derivatives [19,20,21]. Since ibrutinib has high non-covalent binding affinity for BTK, covalent degraders based on this ligand will fundamentally be expected to reduce potency due to loss of catalytic function [20]. Similarly, degraders for KRAS^G12C^ have been generated from existing covalent ligands [22,23]. Although these examples are based on ligands that were developed as covalent inhibitors, the concept of covalency on the target side will apply equally to ligands developed specifically as binders for degrader purposes, where a covalent route has the potential advantage of identification of ligands for otherwise un-ligandable targets. If a covalent route is taken for a degrader, additional challenges include development of a selective covalent ligand (POI or E3) as well as characterization of the rate of covalent bond formation compared to the rate of degradation [19]. General considerations for covalent drug discovery beyond the scope of this review have been reviewed elsewhere [24]. 

In contrast to inhibitors, the pharmacology of degraders can lead to a disconnect between pharmacokinetic (PK) and pharmacodynamic (PD) properties. This is, in part, related to the re-synthesis time of the target, where slowly re-synthesized proteins may remain absent for a significant time after the degrader has been cleared from circulation. Exemplifying a PK/PD disconnect, degraders for the protein RIPK2 show a prolonged functional response after dosing [25]. This is an advantage shared with covalent inhibitors; however, degraders have the additional benefit of catalytic functionality. 

A further advantage with degraders is that they are not required to bind to an active site in a target. This opens up the potential for developing therapies where the target active site is either un-ligandable or absent. In these situations, identification of a small-molecule binder for the target protein is required, which may still pose a considerable challenge. 

The parameters to assess in vitro potency of degraders in drug discovery approaches are principally DC_50_ and D_max_, which correspond to the compound concentration at half-maximal degradation and the maximum degradation, respectively. More comprehensive compound characterization at the screening stage may also include assessment of the rate of degradation [26] and ternary complex formation (vide infra).

## 2. Ubiquitin Ligases in TPD Drug Discovery 

Ubiquitin ligases, or E3s, mediate the post-translation modification of substrates with the small protein ubiquitin. Iterative ubiquitination of ubiquitin itself leads to protein poly-ubiquitination, which mediates different downstream functions depending on the ubiquitin chain architecture; largely, lysine 48 poly-ubiquitinated proteins are degraded by the 26S proteasome [27]. 

Out of the approximately 600 ubiquitin ligases in the human proteome, only a small number have currently been explored in TPD approaches, principally due to the paucity of existing E3 ligands. The most utilized ligases are CRL4^CRBN^ and CRL2^VHL^, where bifunctional degraders based on both have entered the clinic. Other E3s that have been demonstrated to function in a degrader approach include the non-cullin–RING ligases MDM2, IAPs, RNF4 and RNF114. Cullin–RING ligases (CRLs, vide infra) utilized for TPD include Ahr, KEAP1, DCAF1, DCAF11, DCAF15, DCAF16, FEM1B, and KLHDC2 [28,29,30,31,32]. While no experimental evidence is available for use of HECT ligases in degrader approaches at this time, UBR5 was recently suggested as an interesting TPD candidate for this class of E3s [33].

An interesting concept is the identification of essential ligases, which may reduce the occurrence of resistance mechanisms due to loss of ligase expression or mutations that reduce ligase activity. The DCAF1 ligase is reported to be essential and has recently been demonstrated to be active in the context of bifunctional degraders [32]. 

Utilizing ligases that are critical or upregulated in a certain disease condition may allow for effective target degradation primarily in affected tissues, for example in oncology [34]. To enhance tumor specificity, bifunctional degraders have been coupled with antibodies recognizing specific tumor types. Antibody–drug conjugates (ADCs) are an interesting approach to reduce side effects when using degraders that recruit ubiquitously expressed ligases [35]. 

Similarly, taking advantage of differential tissue expression of E3s may allow preferential degradation in the tissue that is relevant to the disease, and an early example of this is the BCL-XL degrader DT2216, which has reduced toxicity due to low VHL expression in platelets [36].

Important aspects to consider when developing a degrader approach is the tissue expression profile and subcellular localization of the E3. For a degrader to be successful, the target and ligase are required to be present in the same cellular compartment, and for clinical application, the ligase must be expressed, and active, in the tissue that is relevant for the indication. VHL and CRBN are widely expressed and have been reported to degrade both nuclear and cytosolic targets [37]. Other ligases may have activity that is restricted to a certain compartment in the cell, such as DCAF16, reported to be limited to the nucleus [38]. Plasma membrane bound ubiquitin ligases, such as RNF43, have also been shown to be possible to hijack for TPD purposes, which opens up an interesting prospect of targeting transmembrane proteins for degradation via the use of degraders that act extracellularly [39,40]. 

While the vast majority of characterized ubiquitin ligases transfer ubiquitin to lysine residues, E3s that act on serine and threonine have been described and could in principle be used in TPD approaches [41], and although there is no current evidence to support this, it is interesting to speculate if this may open up opportunities for degrading proteins that may be recalcitrant to induced degradation, e.g., due to the positioning of lysines in the ternary complex. 

## 3. Ternary Complex Assessment for Compound Development

The activity of both bifunctional and molecular glue degraders is reliant on the formation of a productive ternary complex, where the compound is bridging the ubiquitin ligase with the target protein in a spatial orientation that is compatible with transfer of ubiquitin to lysines in the target protein. It is worth noting that formation of a ternary complex does not necessarily translate into ubiquitination and degradation [42,43] as the target protein, while recruited successfully to the ligase, may not be positioned to allow ubiquitin transfer to lysines. 

Ternary complex assessment can be performed both in cell-based assays or using purified proteins and can provide both qualitative and quantitative information that can aid compound development. An emerging key parameter of ternary complex formation is cooperativity. Cooperativity, denoted as α, measures how the affinity of the degrader for the ligase is affected by simultaneous binding to the target, or vice versa, and is defined as the binary affinity divided by the ternary affinity [44] (Figure 1). Cooperativity can be positive, non-cooperative, or negative. Positively cooperative degraders bind with higher affinity to the ligase in the presence of the target protein and α is >1. Conversely, if the presence of the target protein decreases the binding affinity to the ligase, α is <1, and the degrader exhibits negative cooperativity. Compounds that are not affected in either direction, with α = 1, are termed non-cooperative. Molecular glues typically have high positive cooperativity with α > 1, while bifunctional compounds range from negatively to positively cooperative.

Positive cooperativity suggests that the ternary complex is stabilized by energetically favorable de novo interactions between the ligase and the target protein [45] but may also derive from more unexpected sources, as recently exemplified by Geiger et al. [46]. Here, the authors describe a potent VHL-based degrader for FKBP51, where the VHL and FKBP51 ligands act in a molecular glue-like manner with their respective non-partner protein. In a similar manner, linkers have been shown to contribute to ternary complex formation [47,48], and optimization of linker–protein interactions can enhance cooperativity, as shown for SMARCA2 degraders [47]. 

Negative cooperativity may be due to steric clashes that are counterproductive to complex formation, or to other properties of the compounds that shift the equilibrium towards binary complexes. For example, an FKBP12 degrader with negative cooperativity was recently described [46]. This compound (6a2) showed high binary affinity for FKBP12 and VHL, respectively, however, bound dramatically less well to VHL in the presence of FKBP12. Structural studies revealed that the compound collapses onto itself, forming intramolecular contacts in a horse-shoe shape while bound to FKBP12. In this binary complex, the VHL ligand forms contacts with the FKBP12 protein that likely further stabilize the folded conformation, thus preventing the formation of a ternary complex with VHL. 

Importantly, cooperativity has been reported to correlate with increased potency of degradation [44], and a recent study by from Amgen [49] has illustrated that both ternary complex affinity and cooperativity contribute to the potency of bifunctional degraders. This study assessed VHL-based degraders for SMARCA2 and BRD4 using surface plasmon resonance, and the results highlight that cooperativity is highly correlated to potency of degradation. The authors note that although ternary complex affinity and cooperativity are connected, they are not equivalent. Ternary complex affinity reflects the cumulative interactions in the ternary complex, involving all three components. Cooperativity characterizes the impact of a binding partner, such as the E3 ligase, on the interaction between the PROTAC and its other binding partner, the target protein, as defined above. As such, it is for example in principle possible for a PROTAC to display positive cooperativity with a relatively weak ternary affinity, if the binary affinity is weaker than the ternary. Conversely, a PROTAC with high ternary complex affinity may display negative cooperativity if the binary affinity is higher than the ternary. 

In contrast, ternary complex half-life, i.e., the stability of the induced complex between the ligase and target protein, showed target-dependent correlation with potency of degradation. While BRD4 degraders with more stable complexes were more effective degraders, this correlation was weak for SMARCA2 degraders, highlighting the challenges in building predictive models for bifunctional degraders. 

Of note, bifunctional degraders that are non-cooperative or have negative cooperativity may still be highly potent [50,51], calling into question the value of cooperativity as a key parameter in early degrader discovery. However, highly cooperative lead compounds may be advantageous starting points, as they likely will be more permissive to structural changes that optimize DMPK and physicochemical properties at the cost of reduced ligand affinities. As such, a compound series with strong cooperativity may make room for the medicinal chemist to balance potency with properties that progress the compounds towards the clinic. 

A well-described property of bifunctional degraders is their capacity to form binary complexes with the target protein and the ligase at saturating concentrations, leading to a decrease in activity—this generates the characteristic “hook” appearance in a concentration-response curve (Figure 1B). Note that the hook effect is not observed for MGDs, since these compounds do not display measurable affinity to at least one of the two involved proteins. Highly cooperative bifunctional degraders have lower propensity to form binary complexes and a reduced tendency for hook effect at high compound concentrations, which is an advantage for clinical development. 

## 4. Directly Recruiting E3s for TPD

Engaging E3 ligases is a critical step in degraders’ mode of action. Although three groups are now recognized, E3 Ubiquitin ligases were classically divided into two distinct classes based on their conserved hallmark catalytic domain and ubiquitin transfer mechanism: really interesting new gene-type E3s (RINGs) and homologous to the E6AP carboxyl terminus-type E3s (HECTs) [52].

While RING family ligases catalyze a direct ubiquitin transfer from the E2–ubiquitin complex to a substrate protein [53], HECT E3s initially transfer ubiquitin to an internal catalytic cysteine residue before transferring it to a protein [54]. A third group: RING-in-between-RING (RBR) combines features of both RING and HECT families, the N-terminal RING domain first recruits E2–Ub complex, then transfers ubiquitin onto another RING domain, before transfer to the substrate protein [55,56] (Figure 2A). RINGs represent the most prevalent class of human E3 ligases, with approximately 600 members, surpassing the numbers of HECTs (28 members) and RBRs (14 members) [57]. 

To date, only RING ligases have been leveraged in TPD approaches [34]. Members of this family are subclassified into two categories: cullin–RING ligases (CRLs) and non-CRL ligases. CRLs are modular protein complexes consisting of four variable subunits: a RING box protein (RBX1 or RBX2), a cullin (CUL) protein, an adaptor protein and a substrate-recognizing receptor (Figure 2B). CUL proteins are scaffolds that bind an RBX protein through its C-terminus, while its N-terminus binds to an adaptor protein linked to a substrate receptor [58]. 

CRLs are classified based on the cullin subunit used. The mammalian family comprises eight members: CUL1, CUL2, CUL3, CUL4A, CUL4B, CUL5, CUL7 and CUL9 [59], each with their cognate substrate adaptor and receptor proteins. For instance, CRL1 is a complex consisting of a variable component F-box protein, as adaptor protein, and three invariable components: the S phase kinase-associated protein 1 (SKP1), CUL1 and RBX1 [60].

CRL2 and CRL5 ligases recruit both elongins B and C as adaptor proteins while variable suppressors of cytokine signaling (SOCS) box proteins are used as substrate receptors [61,62]. CRL2 is important in a TPD context, due to the ligase complex based on the Von Hippel-Lindau (VHL) substrate receptor. VHL and SOCS-box proteins contain a common motif, known as the BC-box, essential for binding to the elongin BC complex (Figure 2B) [62]. 

A single bric-a-brac-tramtrack-broad complex (BTB) protein is necessary for linking the CUL3 domain to a substrate protein [63,64]. 

CRL4A and CRL4B are highly conversed (82% identity) [65], and both recruit DNA damage-binding protein 1 (DDB1) as adaptor and DDB1- and CUL4-associated factor (DCAF) proteins, such as CRBN, as substrate receptors (Figure 2B) [66,67]. 

Atypical CRL7 recruits a single F-box protein: F-box and WD40 domain 8 (FBXW8) as substrate receptor and SKP1 as adaptor [68]. The substrate adaptors for another atypical cullin family member, CUL9, also described as p53-associated parkin-like cytoplasmic protein (PARC), are not known [68]. 

CRLs are highly dynamically regulated, activated or decommissioned depending on substrate availability. Both processes depend on the small ubiquitin-like modifier NEDD8, nearly 60% identical to ubiquitin but with distinct targets and functions. Covalent modification of a C-terminal lysine in cullins with NEDD8 induces a structural change stabilizing an open active conformation facilitating the access of the substrates for ubiquitination. Conversely, deneddylation mediated by the constitutive photomorphogenesis 9 (COP9) signalosome (CSN) enables the exchange of substrate receptors from CRLs via the scaffold protein CAND1 [69].

While CRL are multi-subunit E3s, non-CRL ligases are single-subunit entities. They may have a U-box domain instead of a RING domain, which shares a similar structure but lacks the characteristic coordination of zinc ions. Both domains serve the same purpose in facilitating the transfer of ubiquitin from the Ub-E2 conjugated to the substrate protein. Non-CRL ligases can exist in different forms—either as monomeric, homodimeric, or heterodimeric, with the latter being specific to RING-containing non-CRLs only [70].

The first E3 ligands were designed by mimicking specific peptide motifs in E3 substrates, referred to as degrons [28]. Some degrons are linear sequence motifs, whereas structural degrons depend on the spatial conformation of the folded protein, or on a conformational change generated by a post-translational modification [71]. For example, PROTAC-1 developed by Crews and Deshaies in 2001 recruits the F-box protein β-TRCP associated with the CUL1-RBX1-SKP1 E3 complex through an IκBα phosphopeptide [1]. 

Similarly, three years later, the substrate receptor VHL of the CUL2-RBX1-ElonginB-ElonginC complex, known as CRL2^VHL^, was successfully recruited using the seven amino acid ALAPYIP sequence derived from the HIF-1α substrate [72]. However, these early peptide PROTACs had moderate binding affinity and low cell permeability, resulting in a weak activity. Based on the interaction of HIF-1α with VHL, high-affinity synthetic VHL ligands, such as VH032 and later VH298, were developed via considerable efforts of the Ciulli group, paving the way for successful CUL2^VHL^-based PROTACs [71,73,74]. It is worth noting that while these ligands have high affinity for VHL, their physicochemical properties make them challenging for use in the context of generating orally bioavailable bifunctional degraders. 

Hiroshi Handa and collaborators discovered that thalidomide and its analogues, referred to as immunomodulatory imide drugs (IMiDs), more recently known as Cereblon E3 ligase modulators (CELMoDs), bind to CRBN, the substrate receptor of CUL4–RBX1–DDB1–CRBN complex identified as CRL4^CRBN^ [75]. 

A highly conserved hydrophobic pocket on CRBN allows binding of thalidomide and its derivatives, forming a new molecular surface capable of novel protein–protein interactions (PPIs) and thus recruitment of neo-substrates for degradation [76]. Initial characterization of thalidomide, lenalidomide and pomalidomide showed degradation of the transcription factors IKZF1 and IKZF3 [77,78]. Further work by Ebert et al. showed that lenalidomide also led to the degradation of casein kinase 1 alpha 1 (CK1α), but that thalidomide had no effect on CK1α levels [79], demonstrating that changes around the IMiD core leads to different neo-substrate target selectivity. 

Molecular glues can also trigger degradation by promoting protein polymerization; BI-3802, targeting the oncogenic transcription factor B cell lymphoma 6 (BCL6), illustrates this mechanism (Figure 2C) [80]. Binding of BI-3802 to the BTB domain of BCL6 triggers its homodimerization and subsequent higher-order assembly into filaments. BI-3802-induced polymerization enhances the interaction between BCL6 and SIAH1, an E3 ligase that recognizes a VxP motif distal to the drug-binding site [81], leading to accelerated ubiquitination and proteasomal degradation of BCL6. In comparison, previously discovered CRBN-based heterobifunctional BCL6 degraders did not achieve complete BCL6 degradation and failed to induce a significant phenotypic response in diffuse large B-cell lymphoma (DLBCL) [82]. 

By linking a POI warhead to IMiD/CELMoD compounds, molecular glues can be converted into PROTACs [83]. In recent years, a wave of CRBN-based PROTACs has emerged [84]. It is noteworthy that bifunctional degraders based on IMiDs may retain their MGD-dependent ability to trigger the degradation of Ikaros (IKZF1), Helios (IKZF2), Aiolos (IKZF3) or other targets [85]. This property has been exploited by Nurix Therapeutics with their combined BTK and IKZF degrader NX-2127, currently in Phase 1 clinical trials. 

## 5. Recruiting Alternative UPS Components for TPD

Most of the approaches used for degraders so far have focused on the recruitment of E3 ligases, particularly on the substrate receptors of CRLs, such as VHL, CRBN or DCAF15. As mentioned above, substrate receptors are interchangeable subunits which are docked to the scaffold cullin through an adaptor protein. Hijacking and reprogramming CRL activity is an effective strategy used by several viruses and there is evidence of direct association of different viral proteins with the adaptor protein DDB1, which induces the degradation of host proteins such as the STAT1 transcription factors. DDB1 contains three β-propeller domains, the BPB domain docks the N-terminal of the cullin scaffold, whereas domains BPA and BPC bind DCAFs proteins. Some virus proteins have evolved to functionally mimic DCAFs and bind to the adaptor protein DDB1, thus, recruiting E3 ligase activity to direct the degradation of factors relevant to the immune response and escape its effects, as occurs with hepatitis B virus X protein [86] among others. 

Learning from these viral molecular strategies, it has been proposed that molecular glues could mimic viral hijacking of CRL components. This concept has been demonstrated for the E3 component DDB1, where molecular glue degraders rely on the activity of the CRL4B ligase, but is mechanistically independent of substrate receptors, with MGDs acting directly via the DDB1 adaptor to induce POI ubiquitination [2,4] (Figure 2D). Recently, covalent ligands for DDB1 and another CRL substrate adaptor, SKP1, have been identified and successfully used to generate bifunctional degraders, further establishing that induced degradation is possible via a cullin adaptor protein [87,88]. As mentioned above, cullin–RING E3 adaptor proteins are associated with various substrate receptors, and consequently, these proteins are essential for cell viability, as their removal or dysfunction would lead to impaired functions across a broad range of E3 ligases. As such, exploiting CRL substrate adaptors to build degraders may lessen the probability of resistance mutations that render degraders inactive.

Although ligase recruitment is currently the most well-explored modality to induce target degradation, other components of the ubiquitin proteasome system (UPS) can also be harnessed. Given that E3 ligases depend on the action of E2 conjugating enzymes, some research groups have sought small-molecule binders for E2s. Very recently, by library screening and chemoproteomic approaches, King and collaborators discovered a covalent molecular glue degrader, EN450, that impairs leukemia cell viability in a NEDDylation and proteasome-dependent way. EN450 was found to target an allosteric cysteine on the E2 UBE2D and induced the proximity of the E2 with the transcription factor NFKB1 to induce its degradation in leukemia cells [89]. Shortly after, with this approach, the same group demonstrated that core proteins within UPS machinery, such as E2s, can be exploited for PROTAC application, where the compound EN67 acts as a covalent recruiter for the E2 UBE2D [90] (Figure 2D).

Throughout the development of PROTACs and molecular glues, it has been observed that a limitation is that non-native E3 substrates must be efficiently ubiquitinated by the selected ligase. The efficiency of degradation of the POI is determined by the ubiquitination, and this in turn depends on the existence and availability of a ubiquitin acceptor site on the POI. To overcome these constraints, an alternative strategy would be targeting proteins directly to the 26S proteasome, and thus in principle skipping the requirement for ubiquitination. Bashore et al. explored this strategy, sampling a wide range of chemical space by using mRNA display technology. They found a macrocyclic ligand against the 26S subunit PSMD2. By synthesizing heterobifunctional molecules that bind the proteasome subunit on one side and the protein BRD4 on the other, they were able to successfully degrade BRD4 proteins; they referred to these compounds as chemical inducers of degradation (CIDEs; Figure 2D). To confirm that this strategy avoids CUL3 ligase (CRL3) mediation through the KLHL15 adaptor protein, the authors generated KLHL15-knockout clones using CRISPR-Cas9 protocols. Their results showed that degradation occurs independent of this E3 ligase through direct recruitment to the proteasome [91]. This direct-to-proteasome approach has been further explored by the Ciulli group and others, with induced degradation achieved by leveraging ligands for proteasome associated components such as UCHL5, RPN11, USP14 and RPN13 [92,93,94,95], as well as modified proteasome inhibitors [96], and it will be interesting to see how this strategy compares to ligase recruitment in terms of target scope and efficacy.

## 6. TPD as a Therapeutic Strategy

A fundamental question in a TPD drug discovery campaign is whether the target and indication are suitable for a degrader approach. Arguably, many therapeutically relevant targets may be effectively addressed by adopting a traditional occupancy-based strategy, in particular for enzymes, with a likely faster and more straightforward development route. When considering a degrader approach, it is also worth noting that targets that are rapidly resynthesized may yield less therapeutic benefit, compared to slowly resynthesized proteins. However, there are circumstances where a degrader strategy is advantageous, and these are discussed below.

### 6.1. Selectivity

Selectivity of inhibitors against related proteins is sometimes required to limit toxicity, but not possible to achieve due to high active site or ligand binding pocket homology. Interestingly, degraders offer two potential routes around selectivity challenges. The first relies on identification of a novel ligand that binds the desired target in a domain outside of the conserved areas, and thus will only degrade the intended target. This requires significant time and financial investment, and the success will rely on the nature of the ligand and its suitability for use in bifunctional compounds. Similarly, a molecular glue degrader approach may be investigated limiting the glue to a non-conserved domain; however, MGD identification is currently challenging. 

The second route takes advantage of the observation that bifunctional degraders based on existing non-selective target ligands can exhibit selective degrader activity. For example, the non-selective ATP-competitive fibroblast growth factor receptor (FGFR) inhibitor BGJ398 was utilized to generate a selective FGFR1/2 degrader, DGY-09-192, sparing FGFR3/4 [15], and selective degraders for STAT3 and STAT5, respectively, have been generated from relatively non-selective ligands [48,97]. Selectivity in this context likely stems from structural differences that dictate how a target interacts with or can be accessed by the ubiquitin ligase, such that ternary complex formation is either unproductive or does not take place efficiently for a subset of proteins within a group ligandable by the degrader. From this argument follows that different ubiquitin ligases may produce different selectivity profiles, as indeed shown for CRBN and VHL in large-scale proteomics studies using promiscuous kinase ligands [42]. Another example of ligase-driven selectivity is the use of the RNF114 ligase with the non-selective tyrosine kinase inhibitor dasatanib as target recruiter. Although linker chemotype and exit vector geometry will likely impact selectivity, data indicate that VHL and CRBN favor c-ABL degradation, whereas RNF114 preferentially drives degradation of BCR-ABL [98,99]. The expansion of ligases validated for TPD approaches will undoubtedly provide further opportunities to achieve ligase-driven selectivity based on use of non-selective target ligands.

### 6.2. Degraders for Non-Enzymatic Functions

Degraders may provide a way to develop effective drugs for proteins driving biology through scaffolding functions or protein–protein interactions (PPIs), instead of enzymatic activity. TPD is seen as a promising approach for challenging targets like transcription factors, scaffolding proteins, and receptors. However, despite this potential, addressing traditionally undruggable targets with degraders presents significant challenges, with a key consideration being whether identifying a novel ligand for the target is feasible. 

Many clinically relevant enzymes have been attempted to target using inhibitors, and existing ligands can be leveraged to build degraders that address both enzymatic and non-enzymatic functions. Examples here include the IRAK4 and BTK kinases, where both enzymatic and scaffolding functions mediate signaling [100,101,102]. Whereas inhibitors can only address the active-site functionality of these targets, degraders have been demonstrated to be more effective due to removing all functionalities of the targeted protein [103,104]. 

Inhibiting PPIs is generally considered difficult, where interacting proteins often share a relatively large and flat binding interface with little opportunity for ligandable pockets [105]. While the concept of targeting PPIs with degraders has been demonstrated utilizing existing PPI inhibitors for BCL family proteins [36,106], identification of ligands that bind outside of PPI interfaces opens up for addressing a broader variety of PPIs via degraders, and it will be interesting to see if the maturing degrader field will exploit this possibility. 

A highly attractive property of degraders is that they may offer a route for removal of aggregated proteins, a hallmark of many neurodegenerative disorders (discussed in detail below). In this scenario, a degrader strategy can offer clear advantages; however, identification of aggregate specific ligands and access of the degrader to the CNS pose significant challenges. In particular, bifunctional degraders are difficult to optimize for blood brain barrier (BBB) penetration, and as such MGDs are an interesting option for degraders that act within the CNS. 

### 6.3. Resistance Mutations

Resistance mutations in cancer treatments pose a substantial clinical challenge, where cells undergo genetic alterations that confer resistance to initially successful therapeutic interventions, such as upregulation of expression levels or point mutations. In cases where resistance mutations occur in the binding site of small-molecule inhibitors, degraders may offer an advantage. Since degraders do not rely on binding to the active site of the target protein, they can potentially overcome mutations that interfere with traditional inhibitor binding. Moreover, due to their event-based pharmacology and potential for cooperative ternary complexes, degraders may be more forgiving to affinity-reducing mutations in the ligand binding site of the target [107]. 

An example where degraders have been shown to have activity against resistance-associated mutants is BCR-ABL1, present in approximately 95% of chronic myelogenous leukemia (CML) cases. The introduction of imatinib, an ATP-competitive tyrosine kinase inhibitor, marked a significant milestone in treatment of CML [108], which was followed by a number of second-generation inhibitors, including dasatanib [109]. Unfortunately, resistance to both imatinib and second-generation compounds inevitably arise upon treatment. The concept of a degrader approach for BCR-ABL1 has been demonstrated utilizing both ATP-competitive inhibitors and compounds binding allosterically, where bifunctional degraders have activity against treatment-resistant BCR-ABL1 mutants [110,111,112]. In particular, the dasatanib based degrader SIAIS056 was shown to efficiently degrade BCR-ABL1 mutants associated with dasatanib resistance [111]. Likely this is due to the event driven pharmacology of the degrader, where resistance mutations may decrease ligand affinity [113], and consequently reduce efficacy of an occupancy-based therapy, but still allow a significant level of induced degradation. It remains to be seen whether specific BCR-ABL1 degraders will be of value for clinical development to supplement the inhibitor pipeline. 

In a clinical setting, a recent study has focused on BTK in the treatment of B cell cancers like chronic lymphocytic leukemia (CLL). This report characterizes two types of drug resistance mutations in BTK: kinase proficient and kinase impaired. Kinase-impaired mutants, exemplified by L528W, retain downstream BCR signaling despite reduced BTK kinase activity. NX-2127 from Nurix Therapeutics (Table 1) is a potent BTK degrader that is currently in Phase 1 clinical trials, where it has demonstrated over 80% BTK degradation and positive clinical responses in 79% of evaluated CLL patients, irrespective of mutant BTK genotypes, showing promise for overcoming BTK resistance mutations [102].

A further clinical example of where degraders have been used to mitigate resistance mutations is the androgen receptor (AR), an important target in prostate cancer (PC). The AR protein is a transcription factor that promotes cell proliferation when activated by binding to the androgen hormones. In the initial phases of the disease, PC often relies on androgens, and surgical/chemical castration or AR antagonists are employed to deprive the tumor of AR activity [114]. Unfortunately, various mechanisms lead to resistance to androgen-based therapy, including AR overexpression or mutations that convert antagonist activity into an agonist activity, as reported for the clinical AR antagonist enzalutamide [115]. Given the nature of resistance mechanisms for the AR, adopting a degrader approach can provide new routes for treatment options, and pre-clinical examples include the VHL-based AR degrader ARCC-4, utilizing enzalutamide as an AR recruiter. This degrader was demonstrated to effectively degrade the AR F876L antagonist-agonist conversion mutant [116]. Further pre-clinical examples are ARD-61 [117] and MTX-23 [118], where the latter was developed utilizing a ligand for the AR DNA binding domain (DBD), exemplifying the use of ligands that bind outside of the androgen binding pocket to facilitate degradation. ARV-110 (Bavdegalutamide), developed by Arvinas, was the first AR degrader to be evaluated in patients, and currently there are four additional bifunctional AR degraders in clinical trials (ARV-766, CC-94676, AC176 and HP518; Table 1). 

## 7. Discovery Approaches for Bifunctional Degraders

A key requirement for development of a bifunctional degrader is to have access to a ligand for the POI (assuming use of one of the known ligase ligands). Therefore, some of the most important current efforts for development of heterobifunctional degraders include the search for new high-specificity POI binders (Figure 3). 

A second area of great importance for PROTAC development is the search for new molecules recruiting distinct ubiquitin ligases, which is the preoccupation of many public and private laboratories. There are several reasons justifying the need for expansion of the E3 binders’ toolbox. One of the most important aspects is the rapid emergence of resistance, particularly in cancer. If mutations occur in ligases such as CRBN or VHL, or if there is downregulation of the ubiquitin conjugation machinery, current PROTACs may become non-functional [119,120,121]. 

Moreover, it is particularly important to have the choice of using ligases that might give better results in terms of proteolytic activity or if combinatorial treatments are suited. Also, it is important not to hijack all the activity of a ligase, which may block crucial endogenous functions, resulting in unwanted toxicity. These instances underscore the necessity of identifying and developing new binders for alternative ubiquitin ligases. 

### 7.1. Aspects to Consider for POI Ligand Development

As mentioned above, the most common strategy used to block the function of a protein has been the development of small chemical inhibitors to target its active site. This effort resulted in generation of hundreds of molecules specifically recognizing a target, but in many cases failing to inhibit functional activity. 

With a PROTAC strategy, these molecules can be successfully re-employed to build bifunctional degraders. However, when targeting proteins previously considered “undruggable”, new small molecules still need to be specifically designed for degrader development. 

Usually, traditional small-molecule research focuses on developing high-affinity inhibitors targeting an enzyme through its active site to neutralize its effect by an occupancy-driven mechanism. Since heterobifunctional degraders act in an event-driven mechanism, where they only need to interact transiently with their target [122], compounds with moderate POI affinity (≥1–500 nM) may be sufficient. Alongside this, ternary complex cooperativity (vide supra), can allow for relatively weak binary ligand affinity.

Recently, a framework has been created to assess whether human proteins are “PROTACtable”. This system considers the subcellular location of the protein, its ubiquitination site(s), its half-life, and the availability of one or more small-molecule ligands. The workflow is integrated into the Open Targets Platform (https://platform.opentargets.org/, accessed on 7 February 2024).

Despite “PROTACtable” proteins not requiring an enzyme active site, a small-molecule ligand-binding remains necessary. In the case of scaffolding proteins, the selection of a binding site is particularly challenging since the POI might be only partially exposed within a given complex. 

Many currently available degraders are based on existing small-molecule inhibitors, and it is worth noting that inefficient inhibitors can be successfully repurposed to build efficient heterobifunctional degraders, exemplified by ligands developed for TRIM24 [123]. Building upon these early tool compounds, the validation of novel targets using degraders is becoming an increasingly popular strategy in drug discovery.

For targets where there are no existing inhibitors, it is necessary to identify a ligand for the POI. As the only requirement for compounds is binding to the protein in question, screening technologies based on DNA encoded libraries (DEL) or Affinity selection-mass spectrometry (AS-MS) are highly suited for identification of novel ligands for challenging targets, for either a POI or an E3. 

A DEL is a collection of millions to billions of small molecules individually conjugated with unique DNA tags. Using different screening technologies, isolation of compounds that bind the protein target can be achieved, with the DNA tag acting as a barcode, allowing identification of the binding compound. A detailed description of DEL screening is beyond the scope of this review; however, recent excellent reviews by others are available [124,125]. From a TPD perspective, an advantage with a DEL approach is that the DNA attachment point to the library compound may also serve as a linker exit vector. 

Briefly, AS-MS integrates compound binding with mass spectrometric analysis for compound isolation and identification, enabling rapid screening of large compounds collections to identify protein binders [126].

An example of a DEL approach for TPD is the identification of a novel binder for the estrogen receptor α, which was incorporated into a bifunctional degrader [127]. A DEL approach was also utilized to identify a binder of the GID4 ubiquitin ligase [128], which interestingly was also addressed by AS-MS to identify a novel ligase binder [129], exemplifying the use of two different high-throughput technologies to identify structurally different protein binders. 

### 7.2. Aspects to Consider for E3 Ligand Development

The search for new E3 ligases to be used in TPD strategies should consider the ubiquitous (or specific) presence of a ligase in different tissues and cell lines, which may offer benefits in terms of limiting off-target toxicities [36]. As new E3 ligase binders emerge, it will remain important to characterize their scope and applicability, to ensure that the degradation efficiency is at least equal to or better than that observed with VHL or CRBN [23]. Another factor to consider is the subcellular localization of an E3 ligase, since the effect of the degrader could be reduced if its access is limited, for instance, to the nuclear compartment [38].

In addition, several ligases exist in an inactive state or are auto-inhibited in the absence of an activating post-translational modification or binding partner [122]. Using an auto-inhibited ligase may pose additional challenges when considering them for PROTAC development. Naturally, it is also critical that the ligase binder does not inhibit the actual ligase function itself! 

As mentioned above, some ligases are only expressed in specific cell types or are overexpressed in certain pathologies, and this can potentially be leveraged to develop more specific therapies. If the tumor enrichment of an E3 ligase aligns with the dependence of the tumor on the expression of that ligase, this approach could be suitable. In this regard, certain E3 ligases and other UPS components, such as the WD repeat-containing protein 82 (WDR82), have been identified as promising targets essential for various cancer cell types [119]. This dependence may reduce the chances of developing resistance, as observed with some CRBN and VHL-based degraders [119,120,121]. 

In the same vein as for POI ligands, existing ligands for E3s can sometimes be repurposed for use in bifunctional degraders. In the quest to recruit other ubiquitin ligases, the RING ligase mouse double minute 2 homologue (MDM2) was investigated in a TPD approach. Building on the small-molecule PPI inhibitor nutlin-3, which disrupts the interaction of MDM2 with p53, a bifunctional degrader of the androgen receptor was successfully generated [130]. 

To profile novel (or established) ligases in TPD context, approaches include HaloPROTACs and TAG platforms (dTAG and aTAG) [131,132]. These strategies require engineering of a tagged POI, where the tag provides a handle that can be bound by the degrader, allowing evaluation of a ligase across multiple targets via the tag [133]. 

While the lack of binders for both POI and E3 is not a standalone limitation, it does signify the need for additional efforts before initiating the development of a bifunctional degrader. Various approaches, with library screening and structure-based methods being the most prevalent, are employed to identify new binders.

Phenotypic screening is a powerful approach to discover small molecules targeting proteins involved in the regulation of cell physiology and pathology. Several chemical libraries integrating simple or high-complexity molecules have been used by private and public research laboratories [134]. Once integrated into PROTACs, the functionality of the identified molecules as degraders can be screened phenotypically. The chemical properties necessary for drug discovery and the selection of de novo substrates can be defined in the context of phenotypic alterations in cells. Interestingly, this approach can also be used to explore the temporal relationships associated with disease development and response/resistance to treatments.

Rational design of heterobifunctional degraders based on crystal structures and computational chemistry has been used to optimize molecules that will efficiently bind to the POI or E3 ligase. This has also provided information on the formation of the ternary complex, which further allows degrader optimization. Several examples illustrate the success of this approach and have been recently reviewed [134,135]. When crystal structure data are available, this technique can be relatively fast. However, in the absence of such knowledge, two possibilities can be considered: generating the knowledge or using artificial intelligence (AI)-driven tertiary structure prediction models such as DeepMind [136] and RoseTTAFold [137]. These and other approaches become popular if free access exists, like Alphafold from DeepMind, which generates high-quality predicted models of the proteome (https://alphafold.ebi.ac.uk/, accessed on 7 February 2024).

One of the first successful computational workflows for PROTACs development was reported in 2018 [51]. In this work, authors evaluate the linear linkers using a steric scoring scheme. A more recent, practical, in silico tool for PROTAC development considered the contribution of the three components in the ternary complex [138,139]. Since then, several groups have reported tools, some of which are freely available, including the Rosetta-based protocols (https://prosettac.weizmann.ac.il/, accessed on 7 February 2024) [140,141,142]. While some of these methods may be of utility, more work is needed to understand the mechanisms that allow efficient design of heterobifunctional degraders. Artificial intelligence will certainly contribute to better model predictions in the future.

### 7.3. Linker Optimization

The overall degradation efficiency of a heterobifunctional degrader does not simply rely on the affinities of the chemical molecules binding the E3 and the POI. The way in which these two molecules are connected by a linker, allowing the formation of a functional ternary complex (TC) able to induce ubiquitination and degradation of the POI, is as crucial [143]. The length and composition of the linker are critical in the generation of an efficient and specific bifunctional degrader. These design considerations contribute to the efficient formation of a TC [144,145], highlighting their importance for degrader optimization [44,45]. Evidence also clearly indicates the importance of the specific attachment point, or exit vector, to the molecules binding the POI and E3s, which must be optimized for each degrader. Unfortunately, this is not generally predictable due to the structural complexity and dynamics of the TC.

Despite the importance of the linker, well-established strategies for design, resulting in high efficiency, are scarce. Often, stepwise optimization through synthetic alteration of the linker is followed by a degradation activity test, utilizing short and structurally simple alkyl or PEG chains as starting points. Most linkers have consisted of combinations of a few chemical motifs [143,146], the most common being PEG and alkyl chains of varying lengths, which appear in approximately 55% and 30% of linkers, respectively. Approximately 65% of heterobifunctional degraders contain both an alkyl and PEG segment. Only 15% use glycol units, incorporating additional methylene moieties to access different chain lengths. Other less-used motifs include alkynes (7%), triazoles (6%), and saturated heterocycles such as piperazine and piperidine (4% each). Several strategies have been used to improve chemical functionalities of PROTACs, such as photo-switches, conformational locks, and covalent binding. Recent reviews have summarized some aspects of linker chemistry and design strategy [143,147].

To accelerate the screening of bifunctional degraders containing different linkers, direct-to-biology (D2B) approaches, where compounds are assessed for activity in the absence of a purification step, are becoming increasingly popular. This approach allows miniaturization and more rapid cycles of compound testing; however, care must be taken to assess non-specific impact of crude reagent mixtures on cell viability. Using a D2B approach, SAR around linkers, POI ligands, exit vectors and ligase recruiters can be rapidly explored to identify lead compounds for further refinement, and this strategy has been employed by Janssen, GSK and AstraZeneca in recent reports [148,149,150]. 

Critically, linker optimization can make dramatic changes to the overall pharmacokinetic properties of the degrader. Whilst optimization of shape, rigidity, and length impact upon the ability to form a productive ternary complex and, therefore, the degree of degradation observed, these changes can also significantly alter metabolic stability, solubility and permeability. Factors such as aromaticity, 3D shape and polarity deliver subtle changes to the shape and nature of the linker itself but can deliver significantly different behavior in in vivo assays. In addition, the nature of the chemistry used to attach the linker to the POI and E3 ligase binders can also be further optimized, often transitioning from the ubiquitous amide bond (which often limits cellular permeability) to a less problematic linking functionality.

### 7.4. Discovery Approaches for Molecular Glue Degraders

Molecular glue degrader discovery is challenging. Unlike bifunctional degraders, where rational design based on ligase and POI ligands is relatively straightforward, molecular glues modulate protein surfaces to induce interactions in ways that are difficult to predict. 

For this reason, molecular glue discovery has historically been serendipitous, with mechanisms of action identified later. Here, we focus on the discovery of molecular glues which drive novel interactions between a target protein and components of the ubiquitin–proteasome pathways and therefore induce degradation of these target proteins. 

#### 7.4.1. Serendipity

The first known molecular glue degrader, thalidomide, was marketed from the mid-1950s to treat insomnia and morning sickness. At the time, its mechanism of action was unknown and sadly led to the birth of many children with limb defects due to its unidentified teratogenic effects. In 1965, thalidomide and its derivatives, lenalidomide and pomalidomide, were reinvestigated with renewed interest after it was discovered they had immunomodulatory, anti-inflammatory, and anti-tumorigenic properties. However, it was not until 2014 that these immunomodulatory imide drugs (IMiDs) were finally discovered to bind to CRBN, a substrate receptor of the CUL4-RBX1-DDB1-CRBN ubiquitin ligase complex [75,151]. 

A second compound with a mechanism of action analogous to the IMiDs, Indisulam, was originally discovered in the 1990s by screening a library of sulfonamides for cancer cell growth inhibition. Indisulam was shown to cause G1/S cell cycle arrest by flow cytometry and to have in vivo efficacy in human tumor xenograft models [152]. The exact mechanism of action was not elucidated until more than 15 years later, when it was demonstrated that the anti-proliferative effect was due to an induced ternary complex between the mRNA splicing factor RNA binding motif protein 39 (RBM39) and a member of the CUL4 ubiquitin ligase complex called DDB1 and CUL4-associated factor 15 (DCAF15). Formation of this ternary complex leads to the polyubiquitination and degradation of RBM39 [153]. Structural studies showed that indisulam binds DCAF15 and creates a novel surface which enhances RBM39 binding and induces novel protein–protein interactions [154]. 

As molecular glue degrader discovery has historically been serendipitous, it is reasonable to assume there may be small molecules in development or the clinic for which the molecular glue mechanism has not yet been identified. Ebert et al. used database mining to look for a correlation between the cytotoxicity of 4518 clinal or pre-clinical small molecules and E3 ligase expression in hundreds of human cancer cell lines [2]. These studies identified a correlation between CR8 (CDK inhibitor) and DDB1 (CUL4 substrate adaptor protein) expression. An X-ray crystal structure revealed that CR8 bound to CDK12 had a solvent exposed pyridyl moiety which induced complex formation of CDK12 with Cyclin K and DDB1 (PDB:6TD3). This ternary complex formation leads to ubiquitination of Cyclin K and downstream degradation by the proteasome [2]. 

#### 7.4.2. Cellular Screening Approaches

Using cell-based assays to screen for compounds with MGD properties has the potential advantage that the complete cellular machinery is present, allowing ubiquitination and subsequent degradation of a target protein. In contrast, the challenge when utilizing cells for compound screening is the extensive hit deconvolution required for validation of MGD mechanism. 

To help focus this approach, BMS/Cellgene screened a library of CRBN interacting compounds, with the advantage that half the prospective interaction was already defined, and proteomics could be used to identify the target protein. This study was directed towards identification of a molecular glue for the treatment of acute myeloid leukemia (AML), and screening for compounds with antiproliferative potency was performed using ten AML cell lines. This led to the development of CC-90009, a selective GTPS1 degrader, the first CRBN-based molecular glue degrader to enter clinical trials since CRBN was identified to be the primary target of thalidomide [151]. At the time of writing, CC-90009 is in Phase II clinical trials for the treatment of AML (Table 2).

Further expansion of MGDs beyond CRBN requires methods with broader scope. Most of the cullin–RING ubiquitin ligases require conjugation to NEDD8 for activity, and therefore differential screening in cells with normal and deficient neddylation can identify compounds that require uninterrupted neddylation, inferring that activity is driven by a cullin–RING ubiquitin ligase. 

Mayor-Ruiz et al. used this approach to screen 2000 cytostatic or cytotoxic small molecules, with identification of differentially active compounds. The results were followed up by analysis in CRISPR knockout cell lines of all known cullin-RING ligases and quantitative expression proteomics of treated cells to identify the relevant ligase and target protein respectively. This strategy identified three compounds which induced degradation of Cyclin K by the CRL4 ligase complex [4]. Analysis showed that although these compounds are structurally different from CR8 (identified by data mining described above), they act by a similar mechanism of binding CDK12 and recruiting Cyclin K and DDB1 [155]. 

To limit a MGD to a desired ligase, morphological screening approaches using cell lines with different ligase expression levels can be used. Cell painting assays can be used to assess hundreds of parameters including cell viability, morphology and the detection of multiple organelles or cellular components [156], and Ng et al. used this approach in isogenic cell lines expressing different levels of CRBN. A screen of 132 CRBN binders (assessed by a CRBN fluorescence polarization competition assay first) identified FL2-14 as a GSPT2 molecular glue degrader [157]. 

#### 7.4.3. Biophysical Screening Approaches

As a complementary approach to cell-based systems, purified proteins can be used in biophysical assays to measure the induction of a ternary complex between a POI and ligase. This has the advantage of fully controlling the selection of the E3 and the target. Biophysical assays give a direct read out of ternary complex formation and can therefore help build SAR and drive development of compounds with improved molecular glue properties. However, without the cellular context, biophysical assays cannot give information on target protein degradation and any downstream effects. Of importance for biophysical screening approaches is identifying a target and ligase pairing which is suitable for development, with consideration needed for the target and E3 cellular locations and tissue- and disease-specific expression profiles. 

Although uncommon, a known weak compound-induced interaction provides an advantageous start point. Pomalidomide induces an interaction between CRBN and the transcription factor IKZF1 which leads to the degradation of the latter. Novartis further identified a weak interaction between Pomalidomide, CRBN and IKZF2 but this interaction did not lead to IKZF2 degradation. This suggests there is a recruitment threshold which needs to be met before molecular glue induced interactions are strong enough to lead to target degradation. The Novartis discovery campaign was driven initially by interaction assays between CRBN-IKZF2 and CRBN-IKZF1, allowing optimization for selectivity before the interaction threshold for target degradation had been met [158]. Another recent study from Novartis has highlighted the use of biophysical approaches to identify the first VHL-based molecular glue degrader, where the glue induces binding and degradation of cysteine dioxygenase 1 (CDO1) [159]. 

An alternative approach is to take advantage of a known physiological E3 and POI pair by focusing on a mutated POI where the ability of native ligase to bind and ubiquitinate has been lost or impaired. In this scenario, a molecular glue could be utilized to re-establish the interaction between the POI and ligase to restore the normal degradation pathway. β-catenin is an effector protein in the Wnt signaling pathway which normally interacts with the β-TrCP ubiquitin ligase, leading to β-catenin degradation. Mutant β-catenin, found in some colorectal cancers, has an impaired ability to bind β-TrCP, leading to enhanced oncogenic transcription. Nurix Therapeutics developed an FP competition assay using β-TrCP with BODIPY-TMR labelled β-catenin phosphodegron peptides. Discovery was focused on the serine 37 mutant as this is a hotspot for β-catenin mutations. Using an HTS approach, 350,000 compounds were screened through the FP assay to identify enhancers of the β-TrCP and β-catenin peptide interaction. Promising compounds were validated by orthogonal TR-FRET and surface plasmon resonance (SPR) assays [160]. Using native binding E3-POI partners provides the advantage that there is already an interaction surface between the two proteins, with lysine residues available for ubiquitination. It also avoids any issues that may arise from hijacking unrelated E3s to POIs, such as incompatible expression profiles or subcellular localizations. 

Establishing a rational and unbiased screening approach that is broadly applicable for identification of novel molecular glues across different protein pairs is extremely challenging. Significant efforts have focused around redirecting CRBN towards new targets either by expanding around the IMiDs or looking for new chemical matter. Whilst efforts to expand the IMiD chemical repertoire have been successful, with multiple compounds heading towards the clinic, expanding to look at more E3 ligases may yet open up a range of novel target proteins. Phenotypic screening approaches have identified novel glues, but methods require thorough follow up for target deconvolution and hit validation. Exploiting known weak interactions or, in the case of mutant proteins, using its native E3 ligase may provide advantageous starting points due to an already known protein–protein interfaces and lysine residues available for ubiquitination by the E3 ligase. Molecular glues, although challenging to identify, provide an opportunity hijack the protein degradation pathways using compounds with more favorable physicochemical properties than bifunctional molecules. 

### 7.5. Target Validation via Degraders

In addition to direct clinical application, chemical degraders represent a valuable strategy for characterizing the role and importance of a protein in cellular physiology and disease development. Many candidate targets have been selected based on a vital role in the processes they impact, such as oncogenesis, neurodegenerative disorders or inflammation. Validating new targets in a more efficient way represents a crucial step to justify investing efforts in drug development. While transient (siRNA) or permanent (knockout) genetic approaches can be utilized for target validation experiments, these approaches can have drawbacks. Generally, genetic methods, particularly knockout systems, may lead to compensatory mechanisms, such as upregulation of redundant pathways, that may reduce the phenotypic effect related to the absence of the target protein. In contrast, degraders act rapidly, similar to how drugs act on patients, allowing the assessment of how acute depletion of the target affects the relevant readout. There is also a benefit of degrader use to establish pharmacology in situations where catalytic site inhibition does not match the result of siRNA knockdown.

### 7.6. Assays for Validation of Degraders

After developing an initial degrader compound, it is crucial to evaluate parameters for future improvement of critical aspects to obtain an optimal therapeutic molecule. These parameters include binary target engagement, ternary complex formation, efficient polyubiquitination of the target protein, specific proteolysis, pharmacological effects, solubility, stability, and cell permeability.

Various assays can be employed to evaluate physicochemical, pharmacologic, and biologic properties of degraders. This evaluation opens avenues for future improvements of prototype molecules. Techniques like fluorescent polarization [161,162], time-resolved fluorescence resonance energy transfer [51], AlphaLISA [163], surface plasmon resonance (SPR) [44], and calorimetry [47] can be used to assess target engagement and ternary complex formation. Bioluminescence methods like NanoLuc, NanoTag, or NanoBRET are successful for evaluating cellular permeability [164,165].

#### Confirmation of Mechanism of Action

Mechanistic assays can determine whether POI proteolysis is driven by the ubiquitin–proteasome system (UPS) or the autophagy-lysosome system (ALS). A pharmacological approach, such as inhibiting the proteasome (e.g., with bortezomib) or autophagy (e.g., with bafilomycin A), is recommended. Moreover, inhibition of the ubiquitin-activating enzyme using MLN7243 [166] can be utilized to determine ubiquitin dependence. The NEDD8 activating enzyme inhibitor MLN4924 [167] is also useful to validate dependence on a CRL, if appropriate. Bifunctional degraders can also be assessed for mechanism by competition with isolated ligase or target ligands, or by use of non-binding compound analogues for the E3. The approaches above can provide evidence to support that a degrader functions through the expected mechanism of action. 

In addition to western blot analysis, mechanism of action analysis can be performed using homogenous time-resolved fluorescence (HTRF) [168] and AlphaLISA [169]. These represent excellent methods that allow plate-based, high-throughput, compound characterization of endogenous, untagged, POI. This is an advantage, since artificially elevated expression levels and the use of tags can influence POI degradation [165,170]; if such systems are utilized, care should be taken to validate degradation on the endogenous level. 

Direct verification of POI ubiquitylation can be performed by overexposing western blots of cell extracts treated with proteasome inhibitor. Due to rapid deconjugation, POI ubiquitylation is more easily observed using ubiquitin traps (also known as Tandem Ubiquitin Binding Entities) to capture ubiquitylated proteins for detection by western blot or protein arrays [171,172]. 

In addition to confirming that a novel degrader is on-mechanism, it is informative to assess general selectivity. Mass spectrometry can globally evaluate the effect on the proteome of the specific degradation of a POI, ensuring that consequences are restricted to the proteolysis of the target and its known regulated functions [43,173]. Figure 3 summarizes concepts in degrader discovery and validation. 

### 7.7. Therapeutic Areas and Clinical Application

Despite significant anticipated challenges, including metabolic stability, dosing and routes of administration, several degrader projects have already entered clinical evaluation. In line with pre-clinical expectations, oral bioavailability of these large molecules can be challenging to attain. Measured oral bioavailabilities in animal studies, particularly for earlier state derivatives, are generally low, (often in the 3–30% range) [174]. Despite this, the majority of clinically investigated agents have been found suitable for oral dosing. Indeed, for more optimized candidate-stage molecules, bioavailability in mice or rat in vivo PK studies can reach 50–90%. At the current time, little information exists in the public domain to quantify how these bioavailabilities in lower species translate into humans, as the corresponding matched oral/*iv* dosing studies are rarely undertaken in patients. Understanding and resolving these unknown factors is likely to become important as the field matures. From a patient perspective, low oral bioavailability demands a high pill burden, in order to drive sufficient absorption and free drug exposure in the target tissue to deliver therapeutic benefit. Moreover, the potentially large unabsorbed fraction, which may be excreted largely unchanged, has a significant negative impact on the quantities of active pharmaceutical ingredient (API) required for manufacture and, therefore, on cost of goods. Particularly in cases where degraders are competing with small-molecule inhibitors of the same target or pathway, improvements here may be critical to satisfy cost/benefit analyses prior to approval.

#### 7.7.1. Heterobifunctional Degraders in Oncology

As of the time of writing, there are 20 clinical assets in the oncology heterobifunctional space (Table 1). Of these, 70% are dosed orally, and 30% intravenously. One compound (ARV-471) is now in advanced Phase III clinical trials which are not expected to conclude until 2028.

For the majority of clinical degraders, the proteins of interest have already been investigated with significant numbers of small-molecule agents. Examples here include the estrogen receptors (ER) in breast cancer (BC), targeted with selective estrogen receptor degrader small molecules (SERDs), the androgen receptor (AR) in metastatic castrate-resistant prostate cancer (mCRPC), targeted with selective androgen receptor degrader small molecules (SARDs), and kinases such as mtBRAF and BTK, targeted with both covalent and non-covalent inhibitors. As these trials progress, it will be interesting to see where similarities and differences occur in terms of patient responses and outcomes. 

To date, the limited emerging clinical data from those agents progressing beyond Phase I remains relatively modest in terms of response rates. In the case of ARV-471, Phase II data suggested an overall survival benefit of approximately 3-4 months, and median protein degradation of approximately 70% [175]. In the AR setting, in Phase I studies, ARV-110 showed >50% decrease in Prostate-Specific Antigen (PSA) levels in approximately 16% of the patient population and 2/7 partial responses in the evaluable patient cohort [176]. Of note, in patients with known resistance mutations, such as T878X and H875Y, these responses were somewhat higher, suggesting a role for degraders in settings where tumor heterogeneity or tumor evolution has led to therapeutic resistance with small-molecule inhibitors.

Recent reports have suggested that the modest responses in the clinic may be due to a variety of factors [177], including: The aforementioned low bioavailability and, therefore, potentially subtherapeutic exposure at targetEmergent resistance due to loss of function, or decreased expression of the cognate E3 ligaseElevated rates of protein re-expression in response to treatment, counteracting active degradation by the heterobifunctional agent.

It is important to state that these agents are only the first few to report early clinical data, and as the field matures, and collective wisdom and expertise widens, it is likely that our understanding of the degree of bioavailability, the importance of protein re-expression rates and appropriate clinical settings will help to drive improvements in efficacy and, ultimately, patient benefit. So far, many of the targeted proteins seem to have a rapid resynthesis rate measured in a few hours (approximately 3 h and 4 h for the AR [178], and ER [179], respectively). This rapid resynthesis seems to overcome the rate at which the heterobifunctional degraders can eradicate the POI from the cell [177]. Considering this, targeting proteins with a slower rate of compensatory re-expression may become a preferred application for heterobifunctional degraders. Alongside rapid re-expression of the target protein, clinical resistance has also been observed to arise from alterations of the E3 ligase system, or upregulated expression of multidrug resistance efflux pumps [180]. 

Beyond slowly re-expressed proteins, heterobifunctional degraders may offer other opportunities where more traditional modalities have struggled to gain traction. Of specific note in the oncology space, the emerging degraders of the BRM protein offer a compelling instance where a degrader delivers benefit over a small-molecule inhibitor. Based on the concept of collateral lethality [181], experimental studies [182,183] demonstrated that certain lung tumors undergo loss of the SMARCA4 gene, encoding for the BRG1 protein, an essential part of the SWI/SNF chromatin remodeling complex, either by frameshift mutation or epigenetic silencing [184,185]. This leaves this cell population entirely dependent upon the related protein BRM, encoded by the SMARCA2 gene. In this context, a selective BRM inhibitor would be lethal to the ca. 10% of non-small-cell lung cancers (NSCLC) where the SMARCA4 gene is lost or mutated, but well tolerated in healthy tissue due to their unaltered expression. However, due to the high sequence conservation between the two targets and despite extensive efforts, selective BRM inhibitors (for either the bromodomain or ATPase domain) remain elusive.

Here, PRT3789 from Prelude Therapeutics offers clear differentiation from these small-molecule efforts [186]. Biochemically, the heterobifunctional molecule binds at equipotent nanomolar concentrations to both BRG and BRM1, yet delivers 19-fold selectivity in cell-based BRM/BRG1 HiBiT assays. More interestingly, the compound delivers a 720 pM DC_50_ and 94% D_max_ vs. BRM yet spares BRG1 (14 nM DC_50_ and 76% D_max_), delivering selective cell killing in mtSMARCA4/SMARCA4-del cells, but not in SMARCA4-wt cells, with this selectivity translating into matched in vivo xenograft studies. The precise mechanisms underlying this enhanced specificity have not been described but this outcome reflects those observed in the conversion of non-selective kinase inhibitors to selective kinase heterobifunctional degraders [43]. This approach offers the tantalizing prospect of selectively degrading tumor-specific protein homologs, or members of closely related protein families, in a way which is not possible with small-molecule inhibitors. Whilst in its infancy, this paradigm may offer a real differentiator for heterobifunctional molecules, extending the remit beyond those proteins where high quality and effective small-molecule therapies already exist. 

#### 7.7.2. Molecular Glue Degraders in Oncology

The molecular glue degrader landscape in oncology is represented by two predominant molecular classes—the dominant class of MGDs is derived from the phthalimide CRBN recruiters, and the smaller group includes those that are derived from other chemotypes. Thalidomide, and related derivatives, had long been known to be effective in multiple myeloma, but through unknown mechanisms. Significant work by the Tokyo Institute of Technology and Celgene unraveled this mechanism [75,151] and led to the explosion of interest in CRBN-derived molecular glues. As such, thalidomide was the first approved CRBN-molecular glue, well before it was known to act in this manner. Approvals for lenalidomide and pomalidomide have since been granted.

As of the time of writing, a further 16 molecular glues have now entered the clinic (Table 2). Of these, for those where a structure has been disclosed, only three glues (Indisulam, CQS and E7820) do not display the structural motifs common to CRBN molecular glues, instead recruiting DCAF15 to effect protein degradation of RBM39. This overwhelming focus upon CRBN molecular glues raises several challenges, including the race to discover (and patent) novel chemical space. From a clinical standpoint, it also implies that many of the dozens of molecular glues now in pre-clinical development are likely to be competing for the same disease segment, and thus the same clinical trial populations, potentially limiting recruitment and delaying evaluation. Clearly, exploitation of a wider range of E3 ligases in the pursuit of MGDs is commercially attractive and likely to deliver wider patient benefit.

Given their smaller molecular weight and potentially improved physicochemical properties, oral bioavailability remains a key attractive feature of molecular glues, and all the compounds in the clinic which recruit CRBN as the E3 ligase are dosed orally, except for CC-90009 which is dosed via an IV infusion. The reasoning behind this outlier route of administration does not appear to have been publicly disclosed at this time. It is interesting to note that, in addition, all clinical examples of DCAF15 molecular glues are also dosed via an IV infusion, suggesting that oral bioavailability is not necessarily guaranteed for molecular glues outside of the CRBN IMiD-derived agents.

#### 7.7.3. Heterobifunctional Degraders for Inflammatory Indications 

TPD is emerging as a promising approach for the treatment of inflammatory diseases, offering a novel therapeutic strategy with potential advantages. Inflammatory diseases, characterized by abnormal immune responses, may involve dysregulated protein expression contributing to pathogenesis, and a TPD approach is suited to address this by selectively removing disease-associated proteins. Another potential benefit lies in the ability to modulate inflammatory signaling pathways by degrading previously un-druggable proteins. 

An example of these strategies is the development of heterobifunctional degraders for the protein RIPK2. Acting downstream of pattern recognition receptors, RIPK2 activates NF-κB, leading to the production of inflammatory cytokines. Dysregulation of RIPK2 is implicated in various inflammatory diseases, such as Crohn’s Disease and Ulcerative Colitis. Degraders for RIPK2 have been explored pre-clinically as a strategy to improve kinase selectivity and pharmacokinetic properties of inhibitors, and have shown promising anti-inflammatory properties [25,187]. Further examples of targets in inflammatory pathways where degraders have been developed include TYK2 [188] and BTK [189], and bifunctional compounds for both are in pre-clinical development for inflammatory indications by Kymera Therapeutics and Nurix Therapeutics, respectively. 

Currently, the only target where a bifunctional degrader has reached the clinic for an inflammatory indication is IRAK4. As a serine/threonine kinase, IRAK4 functions as a key mediator in the activation of signaling in response to inflammatory stimulation. IRAK4’s central role in inflammatory signaling makes it an attractive target in conditions associated with dysregulated immune activation, and several inhibitors of IRAK4′s kinase activity have entered clinical evaluation [190]. 

Multiple pre-clinical studies of IRAK4 degraders have highlighted that potent degraders for this kinase target can be generated [190], and KT-474 by Kymera Therapeutics is now in Phase 2 for treatment of hidradenitis suppurativa (NCT06028230) and atopic dermatitis (NCT06058156). The Phase 1 clinical results have demonstrated that single dosing of 600 mg–1600 mg KT-474 led to a rapid drop in IRAK4 levels in peripheral blood mononuclear cells (PBMCs), reaching nadir by 48 h. The degradation is sustained for at least 14 days, as IRAK4 levels did not return to baseline for these doses at that time. Multiple dosing over 14 days (once daily) showed that a dose as low as 25 mg led to robust IRAK4 degradation (92% at nadir). It is worth noting that at the 100 mg dose, IRAK4 levels were still significantly below baseline at day 28, some 14 days post-dosing. For both hidradenitis suppurativa and atopic dermatitis, there was an improvement in clinical symptoms after treatment with 75 mg KT-474 daily for 28 days. These clinical responses were either maintained or continued to improve in the two weeks that were evaluated after dosing was halted. Overall, the Phase 1 results are promising, and indicate that longer term treatment can be achieved with a relatively low dose of the compound, and still achieve a strong degradation of IRAK4. In addition to Kymera Therapeutics, Nurix Therapeutics are in the IND enabling pre-clinical phase for their IRAK4 degrader for rheumatoid arthritis and other inflammatory conditions. 

#### 7.7.4. Molecular Glue Degraders for Inflammatory Indications 

At this time, there are no clinical examples of rationally developed molecular glue degraders for treatment of inflammatory disease. However, in the pre-clinical pipeline, Monte Rosa Therapeutics is in the IND enabling phase for an MGD, MRT-6160, that targets VAV1, a protein implicated in T- and B-cell receptor signaling. Both Monte Rosa and Captor Therapeutics are also in the discovery phases for molecular glue degraders directed to NEK7, an activator of the NLRP3 inflammasome.

#### 7.7.5. Heterobifunctional Degraders for CNS Disorders

Whereas targeted protein degradation in oncology has largely exploited targets with matched small-molecule therapeutics, CNS disorders present opportunities for significant differentiation from both small- and large-molecule therapeutic approaches. Misfolded protein aggregates, a hallmark of neurodegenerative disorders, have been considered undruggable using conventional inhibitor approaches, potentially accounting for the failure of compounds in clinical trials targeting protein aggregates in the CNS [191]. 

At this time, a large number of CNS disorders lack effective treatment, and heterobifunctional degraders represent an attractive therapeutic instrument due to their ability to remove proteins via proteasomal degradation. Despite their potential for degrading a POI in vitro, a major challenge for heterobifunctional degraders is their ability to reach the brain and treat the disease in vivo. Due to their high molecular weight and a large polar surface area, achieving blood brain barrier (BBB) permeability of heterobifunctional degraders is particularly challenging. In this context, degrader–antibody conjugates or encapsulated nanoparticles that can cross the BBB through receptor-mediated transcytosis, have been investigated and may provide an alternative route to access the CNS [192,193].

Ubiquitin ligases may have tissue-specific expression, such as RNF182, expressed preferentially in the brain; CNS-restricted ligase expression enables the potential for a more specific targeting of the POI within the CNS [122,194], with the possible advantage of limiting unwanted effects in tissues not directly involved in disease pathology.

Unlike the oncology exemplars described above, PROTACs for neurodegenerative disorders remain in pre-clinical development, largely given the challenges of targeting the brain. Nevertheless, steps have been taken to advance PROTAC molecules in the neuroscience field, with a particular focus on Alzheimer’s disease (AD), Parkinson’s disease (PD), Huntington’s disease (HD), frontotemporal dementia (FTD) and amyotrophic lateral sclerosis (ALS).

##### AD

The accumulation of mutated tau species into neurofibrillary tangles, both in the intracellular and extracellular space, leads to cellular toxicity and neuronal cell death [195]. Different peptide-based and small-molecule-based PROTACs have been developed for tau degradation. 

TH006, a peptide-based PROTAC with an ALAPYIP sequence as a VHL ligand, was shown to induce tau degradation in N2a cells overexpressing tau. Whilst the effect of degradation was observed at high concentrations (200 μM), TH006 administration in a mouse model of AD resulted in a reduction in tau levels in the cerebral cortex and hippocampus. TH006 has been administered both intranasally and intravenously, probably due to low BBB permeability, a common feature of these peptides [196]. Similarly, a Keap1-based peptide PROTAC was observed to recruit CUL3^KEAP1^ ubiquitin ligase, inducing the degradation of tau through the ubiquitin–proteasome system [197].

QC-01-175, a small-molecule tau-degrading PROTAC based on the PET tracer T807, with a CRBN ligand, was shown to induce Tau degradation in FTD patient-derived neuronal models including the tau-A152T and the tau-P301L variants [198]. Following linker optimization, second-generation CRBN-recruiting degraders (FMF-06-series) delivered a tenfold improvement in degradation potency of total tau and phospho-tau S396 with a tau reduction by 50% at 10 nM concentration [199]. Minimal E3-ligase occupancy was observed in in vitro cellular target engagement assays, indicating low cell permeability of the optimized analogues [199].

C004019, a VHL-based degrader, was demonstrated to lead to tau degradation through the proteasome, both in vitro and in vivo. C004019 has shown a DC_50_ of 7.9 nM in a HEK293-hTau overexpressed cell line. Furthermore, intracerebroventricular or subcutaneous administration of C004019 promoted a sustained tau clearance in vivo [200]. Another example is compound I3, a CRBN-based degrader with a THK5105 derivative as a tau ligand, where tau degradation has been demonstrated in vitro using PC12 cells [201].

In the clinical development pipeline, Arvinas claims to have achieved BBB penetration and removal of 95% of pathological tau in vivo after parenteral administration in animal models; the structure of this PROTAC has not been disclosed at the time of writing.

##### PD

Mutations in the leucine-rich repeat kinase 2 (LRRK2) gene and in the α-synuclein gene (SNCA) are linked to the onset of PD. The dysfunction of LRRK2 may contribute to the accumulation of α-synuclein, consequent microglia activation, neuroinflammation and ultimately neuronal cell death [202]. Selective degraders targeting LRRK2 have been recently disclosed in a patent by Arvinas, claiming the discovery of potent and selective CRBN-based LRRK2 PROTACs able to degrade LRRK2 in the cerebral cortex after single oral administration in a fPD mouse model with G2019S LRRK2 mutation [203,204]. In parallel, the Ciulli lab discovered the VHL-based degrader XL01126 that has been shown to be a potent degrader of LRRK2 in multiple cell lines and to pass through the BBB after oral or parenteral administration in mice, exhibiting an oral bioavailability of 15% [205]. These attractive in vitro and in vivo properties of XL01126 represent an intriguing starting point for further drug development. 

With respect to α-synuclein, a synthetic peptide-based degrader has been shown to degrade intracellular α-synuclein in a recombinant expression system. The degradation of α-synuclein through the proteasome has been shown to rescue mitochondrial defects caused by aberrant α-synuclein accumulation [206]. Moreover, in a recent patent, Arvinas claims the discovery of a series of small-molecule PROTACs, containing a VHL, CRBN, IAP or MDM2 ligand, showing a 65% degradation of α-synuclein at 1 μM in different cell lines [207,208].

##### HD

The aggregation of mutant huntingtin in HD leads to progressive degeneration of neurons in the striatum and cerebral cortex. Small-molecule-based bifunctional IAP-based degraders have been developed showing ubiquitination and degradation of mutant huntingtin in fibroblasts derived from HD patients [209]. Further studies are needed to fully exploit this approach to treat HD, as the degradation activity has been observed to also involve the wild-type huntingtin [210].

##### FTD and ALS

The TAR DNA-binding protein 43 (TDP-43) is an example of a misfolded protein implicated in FTD and ALS. The successful removal of the toxic C-terminal form using a heterobifunctional degrader has recently been reported [211].

Given the challenges of developing new drugs for targeting mutated proteins in the brain, further in vitro and in vivo validation in independent laboratories is required to further explore the viability of heterobifunctional as an effective therapeutic tool to treat neurodegenerative disorders. 

#### 7.7.6. Molecular Glue Degraders for CNS Disorders

In principle, an MGD approach has distinct advantages for the CNS space as compared to heterobifunctional degraders, largely due to the generally much more attractive physicochemical properties of glues. 

At this time, no MGDs have been reported for therapeutically relevant CNS targets. Discovery of MGDs here may favor non-aggregating targets, such as LRRK2 described above; however, glues for aggregating proteins may also be possible to discover, since it has been demonstrated that heterobifunctional compounds can function in this area, e.g., for tau. 

## 8. Concluding Remarks and Future Perspectives

Heterobifunctional degraders and MGDs are both groundbreaking modalities in drug discovery and therapeutics, where both options have gained attention for their ability to selectively degrade disease-causing proteins. A TPD approach offers advantages over traditional inhibition, potentially overcoming drug resistance and providing a more profound and sustained therapeutic effect. 

Moreover, degraders offer a quicker avenue for investigating the molecular mechanisms underlying essential cellular processes in both physiology and pathology, bypassing the need for time-consuming gene-silencing approaches.

The further development of these technologies may pave the way for highly specific and efficient therapeutics, with the potential to address previously undruggable targets and improve the overall efficacy and safety of treatments for different therapeutic areas. 

This review has captured developments in oncology, neurodegenerative disorders, and inflammation. However, there is clear potential outside of these areas, e.g., for fibrotic disease [212], anti-infectives [213,214], and metabolic disorders [215]. As research activities into new discovery approaches, DMPK strategies and novel applications progress, degraders are likely to play a pivotal role in shaping the future landscape of medicine, helping us break bad proteins to alleviate human disease burden.

## Figures and Tables

**Figure 1 cells-13-00578-f001:**
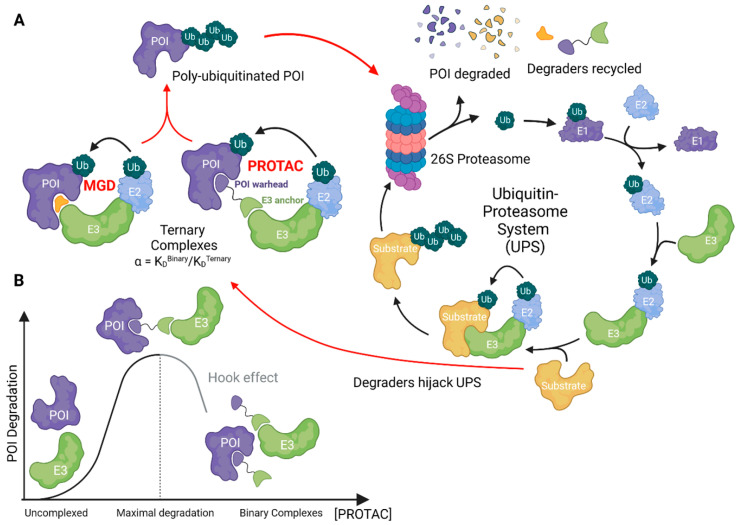
Schematic representation of the mechanisms of proteolysis-targeting chimeras (PROTACs) and molecular glue degraders (MGDs). (**A**) The UPS consists of a cascade involving three enzymes that sequentially attach ubiquitin molecules to target proteins, marking them for proteasomal degradation. Degraders exploit this system by inducing proximity between a target protein (POI) and an E3 ligase, facilitating ubiquitination and subsequent degradation of the target protein. PROTACs are heterobifunctional molecules comprising a POI warhead and an E3 connected by a linker while molecular glue degraders are monomeric molecules. (**B**) Representation of the hook effect illustrating that at high concentrations, bifunctional degraders form binary complexes that reduce target degradation. This figure was created with BioRender.com.

**Figure 2 cells-13-00578-f002:**
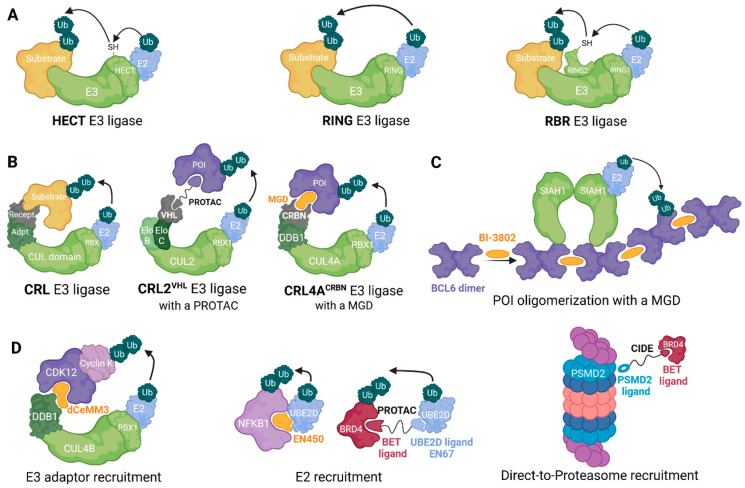
Ubiquitin ligase families and examples of degraders directly recruiting E3 ligases or alternative UPS components. (**A**) The homologous to E6-AP C terminus (HECT) E3 family forms a thioester ubiquitin intermediate with a HECT domain cysteine, before transferring it to the substrate protein. Really interesting new gene (RING) finger E3 ligases utilize a RING domain to facilitate direct transfer of ubiquitin from E2 enzymes to substrate proteins. RING-in-between-RING (RBR) E3 ligases combine features of both RING and HECT ligases; the RING1 domain first recruits the E2–Ub complex before transferring ubiquitin onto a RING2 catalytic cysteine residue, ultimately facilitating its transfer onto the substrate protein. (**B**) The cullin–RING ligases (CRL) are modular RING-family E3 ligases that utilize exchangeable adaptors (adpt) and receptors (recept) organized around a CUL domain to recruit specific substrates. Exemplified are the CUL2-RBX1-ElonginB-ElonginC-VHL (CRL2^VHL^) and the CUL4–RBX1–DDB1–CRBN (CRL4^CRBN^) complexes, with PROTAC and MGD recruiters as indicated. (**C**) MGDs can induce ubiquitination by triggering oligomerization of a protein. For example, MGD BI-3802 induces the dimerization of BCL6, facilitating its recognition by the E3 ligase SIAH1, which then mediates polyubiquitination of BCL6, ultimately resulting in its degradation. (**D**) Degraders can also recruit alternative UPS components to induce POI degradation such as E3 adaptors, E2 enzymes or directly the 26S proteasome. For example, the adaptor DDB1 can be recruited by MGD dCeMM3, forming a ternary complex with CDK12-Cyclin K conjugate, leading to Cyclin K ubiquitination and degradation. The E2 enzyme UBE2D can form a ternary complex with the MGD EN450 and NFKB1, leading to NFKB1 ubiquitination and subsequent degradation. MG EN67 can also be converted into a PROTAC. The proteasome can be directly recruited with compounds such as chemical inducers of degradation (CIDE). These degraders can be used to target a protein of interest (POI), such as BRD4, to the 26S proteasome for ubiquitin-independent proteasomal degradation. This figure was created with BioRender.com.

**Figure 3 cells-13-00578-f003:**
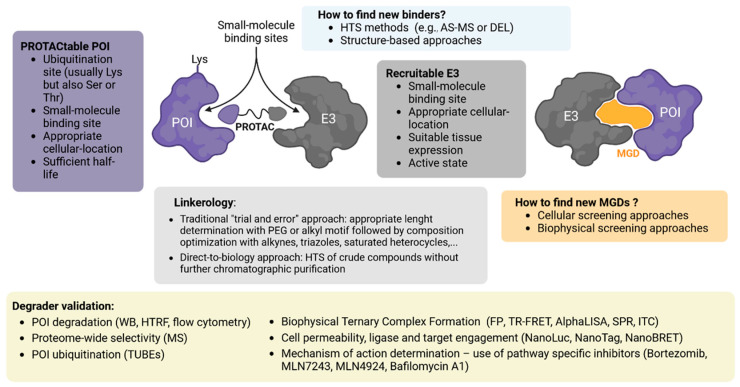
Overview of concepts and considerations for degrader discovery, characterization, and validation. This figure was created with BioRender.com.

**Table 1 cells-13-00578-t001:** Clinically investigated heterobifunctional degraders for oncology indications. Source: https://clinicaltrials.gov, 4 December 2023. Clinical trial identifiers in green denote trials actively recruiting patients. Identifiers in orange are active trials not presently recruiting patients.

Compound	Company	Protein Target	Disease Setting	Dose Route	Current Phase	Clinical Trial ID
ARV-471	Arvinas, New Haven, CT, USA	Estrogen Receptor	BC	Oral	Phase 3	NCT05909397; NCT05654623
ARV-110	Arvinas, New Haven, CT, USA	Androgen Receptor	mCRPC	Oral	Phase 2	NCT03888612
ARV-766	Arvinas, New Haven, CT, USA	Androgen Receptor	mCRPC	Oral	Phase 2	NCT05067140
RNK05047	Ranok Therapeutics, Hangzhou, China	BRD4	DLBCL	iv	Phase 1/2	NCT05487170
BGB-16673	Beigene, Beijing, China	BTK	B-cell lymphomas	Oral	Phase 1/2	NCT05294731; NCT05006716
CFT1946	C4 Therapeutics, Watertown, MA, USA	mtBRAF V600	NSCLC, mCRC, melanoma	Oral	Phase 1/2	NCT05668585
AC176	Accutar Biotech, Cranbury, NJ, USA	Androgen Receptor	mCRPC	Oral	Phase 1	NCT05673109; NCT05241613
ARD-LCC/CC-94676	Celgene/BMS, Lawrenceville, NJ, USA	Androgen Receptor	mCRPC	Oral	Phase 1	NCT04428788
RO7656594	Gemicure/Genentech, South San Francisco, CA, USA	Androgen Receptor	mCRPC	Oral	Phase 1	NCT05800665
HP518	Hinova, Beijing, China	Androgen Receptor	mCRPC	Oral	Phase 1	NCT05252364
DT-2216	Dialectic Therapeutics, Dallas, TX, USA	BCL-xL	Solid tumors/ Hematological tumors	iv	Phase 1	NCT04886622
CFT8634	C4 Therapeutics, Watertown, MA, USA	BRD9	Synovial Sarcoma	Oral	Phase 1	NCT05355753
FHD-609	Foghorn Therapeutics, Cambridge, MA, USA	BRD9	Synovial Sarcoma	iv	Phase 1	NCT04965753
AC676	Accutar Biotech, Cranbury, NJ, USA	BTK (wt/C481S)	B-cell lymphomas	Oral	Phase 1	NCT05780034
NX-2127	Nurix Therapeutics, San Francisco, CA, USA	BTK	B-cell lymphomas	Oral	Phase 1	NCT04830137
NX-5948	Nurix Therapeutics, San Francisco, CA, USA	BTK	B-cell lymphomas	Oral	Phase 1	NCT05131022
AC682	Accutar Biotech, Cranbury, NJ, USA	Estrogen Receptor	BC	Oral	Phase 1	NCT05489679; NCT05080842
AC699	Accutar Biotech, Cranbury, NJ, USA	Estrogen Receptor	BC	Oral	Phase 1	NCT05654532
KT-253	Kymera Therapeutics, Watertown, MA, USA	MDM2	Various tumors	iv	Phase 1	NCT05775406
PRT3789	Prelude Therapeutics, Wilmington, DE, USA	SMARCA2	SMARCA4-del NSCLC	iv	Phase 1	NCT05639751
KT-333	Kymera Therapeutics, Watertown, MA, USA	STAT3	Various lymphoma, leukemia, solid tumors	iv	Phase 1	NCT05225584

**Table 2 cells-13-00578-t002:** Clinically investigated molecular glue degrader for oncology indications. Source: https://clinicaltrials.gov, 4 December 2023. Clinical trial identifiers in green denote trials actively recruiting patients. Identifiers in orange are active trials not presently recruiting patients, those in purple indicate completed studies and those in red are terminated.

Compound	Company	Protein Target	Disease Setting	Dose Route	Current Phase	Clinical Trial ID
Thalidomide (Thalomid)	Celgene/BMS, Lawrenceville, NJ, USA	IKZF1/3	Multiple myeloma	Oral	Approved	
Lenalidomide (Revlimid)	Celgene/BMS, Lawrenceville, NJ, USA	IKZF1/3	Multiple myeloma, Myleodysplastic syndrome	Oral	Approved	
Pomalidomide (Pomalyst)	Celgene/BMS, Lawrenceville, NJ, USA	IKZF1/3	Multiple myeloma	Oral	Approved	
CC-122	Celgene/BMS, Lawrenceville, NJ, USA	IKZF1/3	Melanoma, Hepatocellular carcinoma, Chronic lymphocytic leukemia	Oral	Phase II	NCT03834623, NCT02859324, NCT02406742
CC-92480 (Mezigdomide)	Celgene/BMS, Lawrenceville, NJ, USA	IKZF1/3	Multiple myeloma	Oral	Phase II	NCT05372354
CC-220 (Iberdomide)	Celgene/BMS, Lawrenceville, NJ, USA	IKZF1/3	Multiple myeloma	Oral	Phase II	NCT05199311
ICP-490	InnoCare Pharma, Beijing, China	IKZF1/3	Multiple myeloma	Oral	Phase II	NCT05719701
CC-90009	BMS/Celgene, Lawrenceville, NJ, USA	GSPT1	AML	iv	Phase II	NCT04336982 NCT02848001
CQS	City of Hope Medical Center, Duarte, CA, USA	RBM39	Colorectal cancer, Small-cell lung cancer	iv	Phase II	NCT00005864; NCT00008372
E7070 (Indisulam)	Eisai, Tokyo, Japan	RBM39	BC, AML, Melanoma, Renal cell carcinoma, Colorectal cancer,	iv	Phase II	NCT00080197, NCT01692197; NCT00014625; NCT00059735; NCT00165867, NCT00165854
E7820	Eisai, Tokyo, Japan	RBM39	AML, Colorectal	Oral	Phase II	NCT05024994; NCT00309179
MRT-2359	Monte Rosa Therapeutics, Boston, MA, USA	GSPT1	Myc-driven tumors	Oral	Phase I/II	NCT05546268
CC-99282 (Golcadomide)	BMS/Celgene, Lawrenceville, NJ, USA	IKZF1/3	Non-Hodgkin and B-Cell Lymphomas	Oral	Phase I/II	NCT03930953; NCT04884035
CFT-7455	C4 Therapeutics, Watertown, MA, USA	IKZF1/3	Multiple myeloma, Non-Hodgkin and B-Cell Lymphomas	Oral	Phase I/II	NCT04756726
BTX-1188	Biotheryx, San Diego, CA, USA	IKZF1/3	Non-Hodgkin lymphoma, AML, Solid tumors	Oral	Phase I	NCT05144334
GT-919	Gluetacs Therapeutics, Shanghai, China	IKZF1/3	Hematological tumors	Oral	Phase I	Undefined
GT-929	Gluetacs Therapeutics, Shanghai, China	IKZF1/3	Hematological tumors	Oral	Phase I	Undefined
SP-3164	Salarius Pharmaceuticals, Houston TX, USA	IKZF1/3	Non-Hodgkin lymphoma	Oral	Phase I	NCT05979857
DKY-709	Novartis Pharmaceuticals, Basel, Switzerland	IKZF2	Advanced solid tumors	Oral	Phase I/Ib	NCT03891953

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
