# Peer review of "Breaking Bad Proteins—Discovery Approaches and the Road to Clinic for Degraders"

_cells, 2024, doi:10.3390/cells13070578_

Round 1
Reviewer 1 Report
Comments and Suggestions for Authors
Bouvier et al.
This a well written, very comprehensive review that will be valuable to readers that seek orientation in the field of targeted protein degradation.
In my feeling, the comprehensiveness of the review is also its greatest weakness: it is overlong. I would recommend to remove several sections (at the authors’ discretion).
E.g. the discussion of DNA encoded libraries (lines 565-582) really goes beyond the frame of the review and could be removed. The same applies to lines 611-647, which is very generic and provides little information to experts in the area, but will not suffice to novices orientation.
Also, the discussions in lines 851 to 879 does not reach the level of depth that this review displays in other section. I would propose to delete this part, too.
Line 938, “Anecdotally, this rapid resynthesis seems to overcome the rate at which the heterobifunctional degraders can eradicate the POI from the cell.” This is far from anecdotical, precise models of this have been published, reviewed in (Bartlett and Gilbert 2022).
The discussion of the synthetic lethality of SMARCA2 and SMARCA4 in line 945 to 959 is poor: the referencing misses the original paper (Oike et al. 2013), and the statement that SMARCA4 is a collateral deletion does not reflect the understanding of the field. Also, no reference is provided for this notion. One of the references is a conference abstract, which is not acceptable given the ample literature available here. My advice would be to delete this paragraph, too, since the authors – if you excuse by bluntness – do not seem to be experts in the field.
In line 1174, anti-infectives are proposed as a possible application of PROTACs. I think it would be fair to quote the work of the Clausen lab here, who reported bacterial PROTACs (Hoi et al. 2023; Morreale et al. 2022).
Minor points:
Line 57: “the field is poised to encounter significant challenges...”. “poised” is misleading, the field has been facing these difficulties for years.
Line 64 “dramatically” is an exaggeration and not a scientific term. The frequency of dosing of a drug is influenced by a plethora of factors. Degraders add another factor to the game, but the effect is not “dramatic”.
Line 148 , “…degrading proteins recalcitrant 148 to ubiquitylation on lysines.” I have not seen evidence of any protein yet that is recalcitrant to ubiquitylation on lysines. I would delete the statement, unless a reference can be provided.
Line 193 “complex affinity and cooperativity contribute to the potency of bifunctional degraders”. I find this confusing; cooperativity is defined by the greater affinity of the trimeric complex to binary interactions. So cooperativity and complex affinity are more or less the same thing, to my mind.
Line 213 “these compounds do not bind to the target and ligase individually” does not seem right to me. Glues bind to one of them, e.g. cereblon based glues bind to cereblon. Correct to “these compounds do not display measurable affinity to at least one of the two involved proteins”
Line 320 “Elbert” correct to “Ebert”
Line 556 “Unfortunately, structural information on scaffolding proteins or 556 multimeric complexes is still limited for use as a systematic approach. This is mainly due 557 to the complex architecture of scaffolding proteins, making them difficult to produce in-558 dividually or to integrate into a functional multimeric complex.” This is a broad-brush explanation does not seem convincing to me, and no references are given. I would delete that statement.
Having said all that, I would like to reiterate that this an excellent review that deserves publishing.
Bartlett, D. W., & Gilbert, A. M. (2022). Translational PK–PD for targeted protein degradation. Chemical Society Reviews, 51(9), 3477–3486. https://doi.org/10.1039/d2cs00114d
Hoi, D. M., Junker, S., Junk, L., Schwechel, K., Fischel, K., Podlesainski, D., et al. (2023). Clp-targeting BacPROTACs impair mycobacterial proteostasis and survival. Cell, 186(10), 2176-2192.e22. https://doi.org/10.1016/j.cell.2023.04.009
Morreale, F. E., Kleine, S., Leodolter, J., Junker, S., Hoi, D. M., Ovchinnikov, S., et al. (2022). BacPROTACs mediate targeted protein degradation in bacteria. Cell, 185(13), 2338-2353.e18. https://doi.org/10.1016/j.cell.2022.05.009
Oike, T., Ogiwara, H., Tominaga, Y., Ito, K., Ando, O., Tsuta, K., et al. (2013). A synthetic lethality-based strategy to treat cancers harboring a genetic deficiency in the chromatin remodeling factor BRG1. Cancer Res, 73(17), 5508–18. https://doi.org/10.1158/0008-5472.can-12-4593
Author Response
Dear Reviewer 1, please see the attached pdf document for our point-by-point response.

Reviewer 2 Report
Comments and Suggestions for Authors
The manuscript entitled "Breaking Bad Proteins – Discovery Approaches and the Road to 2 Clinic for Degraders" focusses on drug discovery with respect to bifunctional and molecular glue degraders within the ubiquitin proteasome system, including analysis of mechanistic concepts and discovery approaches, with an overview of current clinical and pre-clinical degrader status in oncology, neurodegenerative and inflammatory disease. The manuscript is well written and well documented.
However, the authors need to address the following points in the manuscript:
1. The manuscript lacks clarity in explaining the specific roles and mechanisms of different types of ligases in targeted protein degradation. It also does not clearly delineate the challenges associated with the development of covalent bifunctional degraders or provide concrete examples to support the points made.
2. What challenges arise during the development of bifunctional degraders concerning their molecular weight and physiochemical characterstics.
3. In what ways do the pharmacological mechanisms of action of degraders differ from those of conventional small molecule inhibitors.
4. What are the pros and cons associated with the development of covalent bifunctional degraders.
5. How do degraders address the discrepancy between pharmacokinetic and pharmacodynamic properties.
Author Response
Dear Reviewer 2, please see the attached pdf document for our point-by-point response.

Reviewer 3 Report
Comments and Suggestions for Authors
The manuscript is well written. The work is logically organized and the topic is comprehensibly described. I have only a few comments.
1. The authors should improve citation of figures in the text. Also, please add panels to specific figures.
Line 37 – Figure 1A
Line 212 – Figure 1B
Line 228 – Figure 2A
Line 248 and 276 – Figure 2B
Line 324 – Figure 2C
Line 358 – Figure 2D
Line 377 – Figure 2D
Line 389 – Figure 2D
2. I think that there is an error on line 235. Figure 2B, not 2A should be referred here.
3. Please, provide explanation of the abbreviation.
Line 898 – API
4. There may be a double space typo in front of “Alongside” on line 941.
Comments on the Quality of English Language
good quality
Author Response
Dear Reviewer 3, please see the attached pdf document for our point-by-point response.
